# PCR bias impacts microbiome ecological analyses

Dharmik R. Rathod[1], Justin D. Silverman [1,2,3,4]*

**1** College of Information Sciences and Technology, Penn State University, University Park, Pennsylvania, United States of America, **2** Department of Statistics, Penn State University, University Park, Pennsylvania, United States of America, **3** Department of Medicine, Penn State University, Hershey, Pennsylvania, United States of America, **4** Institute for Computational and Data Science, Penn State University, University Park, Pennsylvania, United States of America

* JustinSilverman@psu.edu

## Abstract

Polymerase Chain Reaction (PCR) is a critical step in amplicon-based microbial community profiling, allowing the selective amplification of marker genes such as 16S rRNA from environmental or host-associated samples. Despite its widespread use, PCR is known to introduce amplification bias, where some DNA sequences are preferentially amplified over others due to factors such as primer-template mismatches, sequence GC content, and secondary structures. Although these biases are known to affect transcript abundance, their implications for ecological metrics remain poorly understood. In this study, we conduct a comprehensive evaluation of how PCR-bias influences both within-samples ($\alpha$-diversity) and between-sample ($\beta$-diversity) analyses. We show that perturbation-invariant diversity measures remain unaffected by PCR bias, but widely used metrics such as Shannon diversity and Weighted-Unifrac are sensitive. To address this, we provide theoretical and empirical insight into how PCR-induced bias varies across ecological analyses and community structures, and we offer practical guidance on when bias-correction methods should be applied. Our findings highlight the importance of selecting appropriate diversity metrics for PCR-based microbial ecology workflows and offer guidance for improving the reliability of diversity analyses.

## Author summary

PCR amplification is a routine step in microbiome sequencing, but it does not treat all DNA sequences equally. Some bacterial sequences amplify more efficiently than others, creating PCR bias that distorts the relative abundances observed in sequencing data. While this bias is known to affect individual taxa, its impact on widely used ecological diversity measures has been unclear. In this study, we evaluate how PCR bias influences commonly used $\alpha$-diversity (within-sample) and $\beta$-diversity (between-sample) metrics. We show that many popular

reproduction in any medium, provided the original author and source are credited.

**Data availability statement:** All code and data used in this study are available at https://github.com/dharmikrathod/pcr_bias_code.

**Funding:** J.D.S. was supported in part by the National Institute of General Medical Sciences (NIH 1R01GM148972-01). The funders had no role in study design, data collection and analysis, decision to publish, or preparation of the manuscript.

**Competing interests:** The authors have declared that no competing interests exist.

measures–including Shannon diversity, Simpson's index, Bray–Curtis dissimilarity, and Weighted UniFrac–can change substantially due to PCR bias, and that the direction and magnitude of this distortion depend on the underlying community composition. As a result, PCR bias can create or mask apparent ecological differences between groups. In contrast, we identify a class of metrics, such as Aitchison distance and differential log-ratios, that remain unaffected by PCR bias. These findings provide practical guidance on when PCR calibration is important and which diversity measures offer more robust ecological inference.

## 1 Introduction

The emergence of next-generation sequencing technologies, along with advances in computational tools, has substantially expanded microbiome research across biomedicine [1]. Amplicon-based microbiome studies typically target hypervariable regions of the 16S rRNA gene, using them as barcodes to quantify the relative abundance of bacterial taxa in mixed communities [2,3]. To isolate the 16S rRNA gene and generate sufficient material for sequencing, multiple cycles of polymerase chain reaction (PCR) amplification are usually required [4]. However, amplification efficiency can vary across 16S rRNA templates, leading to systematic over- or under-representation of certain taxa in sequencing libraries [5–7]. These biases can lead to over 15-fold errors in observed relative abundances, severely distorting estimates of microbial composition [8]. Overall, PCR bias can be a substantial source of error in 16S rRNA studies [5–11]. Despite efforts to optimize experimental protocols (e.g., [12]), PCR bias remains an outstanding problem.

PCR bias likely originates from multiple distinct processes. Mismatches between primer and template sequences are one source of bias [13]. Yet, this primer-mismatch bias is unlikely to persist past the initial cycles of PCR, after which point the template sequence is replaced by a sequence complementary to the primers themselves [14]. Still other transcript level factors can lead to Non-Primer-Mismatch (NPM) bias which persist through later cycles [8]. For example, GC-rich templates are more stable and can require higher melting temperatures or can form hairpin structures that hamper amplification [15]. Mock community studies suggest that sources of NPM bias dominates primer-mismatch biases [8,11,16]. Because NPM bias is the dominant and persistent source of PCR distortion, we use the terms "PCR bias" and "NPM bias" interchangeably throughout this work.

Recent studies have shown that PCR bias is highly consistent across PCR cycles and can be accurately modeled by a simple exponential bias model. Let $a_1/a_2$ represent the true ratio of two templates before amplification (i.e., at cycle 0), and let $b_1/b_2$ denote the ratio of their per-cycle amplification efficiencies, where each $b_j \in [1, 2]$, with 1 indicating no amplification and 2 indicating perfect doubling. After $x_n$ cycles of PCR, the observed ratio of the two templates becomes $w_{1n}/w_{1n}$. These quantities are related through the following model:

$$\frac{w_{1n}}{w_{2n}} = \frac{a_1}{a_2} \left( \frac{b_1}{b_2} \right)^{x_n}.$$

(1)

This model was originally proposed by Suzuki and Giovannoni [6] who used it to study PCR reactions with two templates. This model was recently generalized by Silverman et al. [8] and McLaren et al. [10] who extended it to the case where there are more than two templates. Silverman et al. [8] further generalized this model to situations where the ratios (e.g., $\frac{w_{1n}}{w_{2n}}$) are not measured directly but observed only through noisy sequencing data. By sequencing 16S rRNA libraries with different numbers of PCR cycles, those authors could directly infer relative PCR efficiencies (e.g., $\frac{b_1}{b_2}$) and the unbiased ratio abundances (e.g., $\frac{a_1}{a_2}$). They found that this simple model explained over 95% of variation due to PCR bias in both mock and real 16S rRNA libraries. Those findings have subsequently been validated by multiple groups [11,16].

Despite advances in modeling PCR bias, relatively little research has investigated which types of microbiome analyses are most affected. For instance, it remains unclear when researchers should perform calibration experiments, such as sequencing replicates with varying numbers of PCR cycles, to enable application of models like Eq (1). Most existing studies have focused on how PCR bias distorts estimates of relative abundance, leading to systematic over- or under-representation of certain taxa. However, such distortions may not affect all downstream analyses equally. For example, both Silverman et al. [8] and McLaren et al. [10] have shown that differential abundance analyses, which aim to estimate changes in relative abundance across conditions (e.g., health vs. disease), can be invariant to PCR bias. In contrast, other commonly used approaches remain under-explored. In particular, ecological diversity measures such as $\alpha$-diversity (e.g., Shannon index) and $\beta$-diversity (e.g., Bray–Curtis dissimilarity) form the basis of many microbiome analyses, yet the extent to which PCR bias affects these metrics has not been systematically studied.

This article presents the first systematic evaluation of how PCR bias affects ecological analyses based on $\alpha$- and $\beta$-diversity metrics. First, we prove that there exists a class of perturbation-invariant diversity measures that are unaffected by PCR bias. Second, we demonstrate that a broader class of perturbation-sensitive metrics—including Shannon diversity and Bray–Curtis dissimilarity—are impacted by PCR bias. Importantly, this bias is not consistent: it depends on the true underlying composition (e.g., $\frac{a_1}{a_2}$), implying that PCR bias can also distort differential diversity analyses aimed at identifying changes in diversity across groups. To guide future research, we provide intuitive explanations of how bias varies with community composition and which types of microbial communities are most susceptible. Based on our findings, we offer a simple recommendation: researchers who adopt perturbation-invariant metrics need not perform calibration experiments to mitigate PCR bias, whereas those using perturbation-sensitive metrics should strongly consider doing so.

## 2 Results

### 2.1 A statistical model for PCR bias

This article uses the model from Silverman et al. [8] which extends Eq (1): accounting for more than two taxa and measurement error. Note that Eq (1) can be extended to $D > 2$ taxa by writing it as a linear log-contrast model:

$$\mathbf{\Psi} \log \mathbf{w}_n = \mathbf{\Psi} \log \mathbf{a} + x_n \mathbf{\Psi} \log \mathbf{b} \tag{2}$$

where $\mathbf{\Psi}$ is a $(D-1) \times D$ contrast matrix with elements $-1 \leq \mathbf{\Psi}_{id} \leq 1$ and rows that sum to zero ($\sum_{d=1}^{D} \mathbf{\Psi}_{id} = 0$ for all $i \in \{1, \dots, D-1\}$) and $\mathbf{w}_n$, $\mathbf{a}$ and $\mathbf{b}$ are all $D$-dimensional, positive-valued vectors that sum to 1 (compositional vectors). For example, a logarithmic version of Eq (1) can be recovered from this linear log-contrast model by letting $\mathbf{\Psi} = (1, -1)$, $\mathbf{w}_n = (w_{1n}, w_{2n})^T$, $\mathbf{a} = (a_1, a_2)^T$, and $b = (b_1, b_2)^T$. For our purposes, we can choose any contrast matrix $\mathbf{\Psi}$ with rank $D{-}1$, e.g., a matrix $\mathbf{\Psi} = (\mathbf{I}_{D-1}; -\mathbf{1}_{D-1})$ which corresponds to the contrast matrix of the Additive Log-Ratio Transform [17]. We use the following notation as a shorthand: $\eta_i = \mathbf{\Psi} \log \mathbf{w}_i$, $\boldsymbol{\alpha} = \mathbf{\Psi} \log \mathbf{a}$, and $\boldsymbol{\beta} = \mathbf{\Psi} \log \mathbf{b}$ resulting in a multivariate linear model: $\eta_n = \boldsymbol{\alpha} + \boldsymbol{\beta} x_n$.

Sequencing itself is a noisy measurement process that produces a zero-laden $D \times N$-dimensional count matrix $\mathbf{Y}$ with element $\mathbf{Y}_{dn}$ representing the number of sequencing reads mapping to taxon $d$ in sample $n$. As a result, we cannot

measure $\eta$ directly, or calculate it directly from $Y$, instead, Silverman et al. [8] proposed the following Bayesian hierarchical model which treats $\eta$ as nuisance parameters to be estimated along with the parameters of interest $\alpha$ and $\beta$:

$$Y_n \sim \text{Multinomial}(\pi_n)$$
$$\pi_n = \phi^{-1}(\eta_n)$$
$$\eta_n \sim N(\alpha + x_n \beta, \Sigma)$$

where $\pi_n$ is a $D$-dimensional vector of relative abundances, $\phi(\pi_n) = \Psi \log \pi_n$ is a $(D–1)$ dimensional vector of log-ratios and $\Sigma$ is a $(D–1)$-dimensional covariance matrix that is also estimated from the observed data. As in Silverman et al. [8], we estimate the parameters in this model ($\alpha$ and $\beta$) and the nuisance parameters ($\eta$ and $\Sigma$) using the *fido* R library which provides scalable Bayesian inference for this type of multinomial logistic normal linear models [18]. This model has been experimentally validated by multiple groups [8,11,16].

When applied to datasets that include calibration samples—i.e., samples sequenced after undergoing different numbers of PCR cycles—this model enables inference of the taxon-specific relative amplification efficiencies $\beta$ and the underlying unbiased relative abundances $\alpha$, with quantified uncertainty. In practice, we often include additional covariates (beyond cycle number, $x_n$) to account for batch effects. By including additional covariates, this model can also be applied to samples from distinct microbial communities, each with their own unbiased relative abundance vector $\alpha$ (See Methods).

To illustrate the model, we reproduce the analysis of Silverman et al. [8] to estimate how PCR bias alters relative abundance estimates in both an *in vitro* and an *ex vivo* human gut microbiome study. S1 Fig, which parallels figures from the original study, visually demonstrates the extent of this distortion. Briefly, some taxa, such as Holdemania, are consistently underrepresented by a factor of approximately 16, whereas others, such as Bacteroides, are overrepresented by a factor of about 4. Overall, many taxa show clear evidence of PCR bias, with 95% credible intervals for their amplification efficiencies excluding the null (no bias).

## 2.2 Sensitivity and invariance of estimands to PCR bias

Most prior work on PCR bias has focused on its effect on relative abundance estimates (e.g., [8]). However, relative abundances are often not the ultimate quantity of interest; instead, they serve as intermediate inputs for downstream ecological or statistical estimands. Here, we show that such estimands can be broadly classified into two categories: *perturbation-invariant* estimands, which remain unaffected by PCR bias, and *perturbation-sensitive* estimands, whose values can change substantially under PCR bias. This distinction provides a principled framework for understanding when PCR bias can be safely ignored and when it must be explicitly accounted for.

Let $\pi_{dn}$ denote the estimated relative abundance of taxon $d$ in the $n$-th sample. A common estimand in differential abundance analyses is the relative log-fold-change: the difference in mean log-ratio abundance between two biological conditions $z_n \in \{0, 1\}$ (e.g., health versus disease):

$$\tau_d = \underset{n:z_n=1}{\text{mean}} \log \frac{\pi_{d_1 n}}{\pi_{d_2 n}} \;-\; \underset{n:z_n=0}{\text{mean}} \log \frac{\pi_{d_1 n}}{\pi_{d_2 n}}.$$

In Corollary 1 (S1 Appendix), we show that $\tau_d$ is an example of a broader class of estimands that are *perturbation invariant*. Informally, an estimand is perturbation invariant if its value does not change when all samples are subjected to the same compositional perturbation–that is, when their log-ratio representations are shifted by the same vector in log-ratio space. Formally, an estimand $\theta$ is perturbation invariant if it satisfies

$$\theta(\pi) = \theta\big(\phi^{-1}\big(\phi(\pi) + \gamma \mathbf{1}_N^T\big)\big)$$

for any $D$–1-dimensional vector $\gamma$ where $\pi$ is a D-dimensional vector of relative abundances and $\phi(\pi) = \Psi \log(\pi)$ is a $(D$–1) dimensional vector of log-ratios.

From the perspective of PCR bias, perturbation-invariant estimands are special because they are also invariant to PCR bias. We prove this formally in Theorem 1 (S1 Appendix). This theorem and corollary formalize prior work which hypothesized that relative log-fold-changes were invariant to PCR bias [8,19]. Intuitively, PCR bias, as modeled in Eq (2), acts as a sample-specific perturbation: it shifts all log-ratio coordinates by the term $x_n\beta$. This shift is analogous to adding a global nuisance term $\gamma\mathbf{1}_N^T$, which uniformly affects all log-ratios and therefore cancels out in any estimand based solely on contrasts (i.e., differences between taxa rather than their absolute log-ratio coordinates). Thus, perturbation-invariant estimands are unaffected by PCR bias. In the following section, we turn to perturbation-sensitive estimands, which do not share this invariance and can be strongly influenced by PCR bias.

### 2.3 Common ecological $\alpha$-diversity analyses are impacted by PCR bias

The use of $\alpha$-diversity metrics in microbiome research has become ubiquitous with the rise of next-generation sequencing (NGS). These metrics provide a quantitative description of within-sample diversity, capturing aspects of richness and evenness, and are central to ecological and microbiological studies. Reliable and reproducible estimates are particularly important because microbial community structure can strongly influence host physiology and ecosystem stability [20]. However, unlike relative log-fold-change estimands ($\tau_d$), we find that $\alpha$-diversity estimands are perturbation sensitive and therefore susceptible to PCR bias.

We evaluated four commonly used $\alpha$-diversity metrics derived from relative abundances: Shannon's index, Simpson's index, the Gini coefficient, and the Aitchison norm using the Bayesian multinomial logistic-normal model described in Sect 2.1 and data from Silverman et al. [8,21–23]. The dataset consists of ten mock communities with controlled proportions of ten bacterial isolates and four human gut microbiota samples from distinct artificial gut systems. All the artificial guts were inoculated with a human fecal sample and cultured ex vivo in a system designed to replicate host physiology. We estimated posterior distributions of each $\alpha$-diversity metric for the true (pre-amplification) compositions and for compositions after 35 PCR cycles (Fig 1). Formally, let $\phi^{-1} : \mathbb{R}^{D-1} \to \mathbb{S}^D$ denote any inverse log-ratio transformation and $f : \mathbb{S}^D \to \mathbb{R}_+$ denote an $\alpha$-diversity metric on the $D$-dimensional simplex. We define PCR-induced bias as:

$$\text{Bias} = f(\phi^{-1}(\alpha + 35\,\beta)) - f(\phi^{-1}(\alpha)),$$

where $\alpha$ represents the estimated log-ratio composition before amplification (0th cycle) and $\beta$ the per-cycle PCR bias.

All four metrics are clearly impacted by PCR bias, with values after 35 PCR cycles substantially differing from their true pre-amplification (0-cycle) values (Fig 1). To further quantify this effect, we also measured the posterior distributions of *relative bias* for each metric in S2 Fig using the ten mock communities and four human gut microbiota samples from Silverman et al. [8]. We define PCR-induced relative bias as:

$$\text{Relative Bias} = \frac{f(\phi^{-1}(\alpha + 35 \cdot \beta)) - f(\phi^{-1}(\alpha))}{f(\phi^{-1}(\alpha))}.$$

Across both figures, the magnitude and direction of the bias varies across communities, reflecting its dependence on the underlying composition. As a result, PCR amplification can distort not only absolute estimates of diversity but also comparisons across experimental conditions–that is, differential $\alpha$-diversity analyses such as assessments of whether diversity differs between cases and controls (e.g., [24]).

To assess how strongly PCR bias could affect differential $\alpha$-diversity analyses, we used mock community data from Silverman et al. [8]. Because the original study did not include natural experimental groupings, we applied an optimization procedure to construct groupings that maximized changes in statistical signal across PCR cycles. We quantified

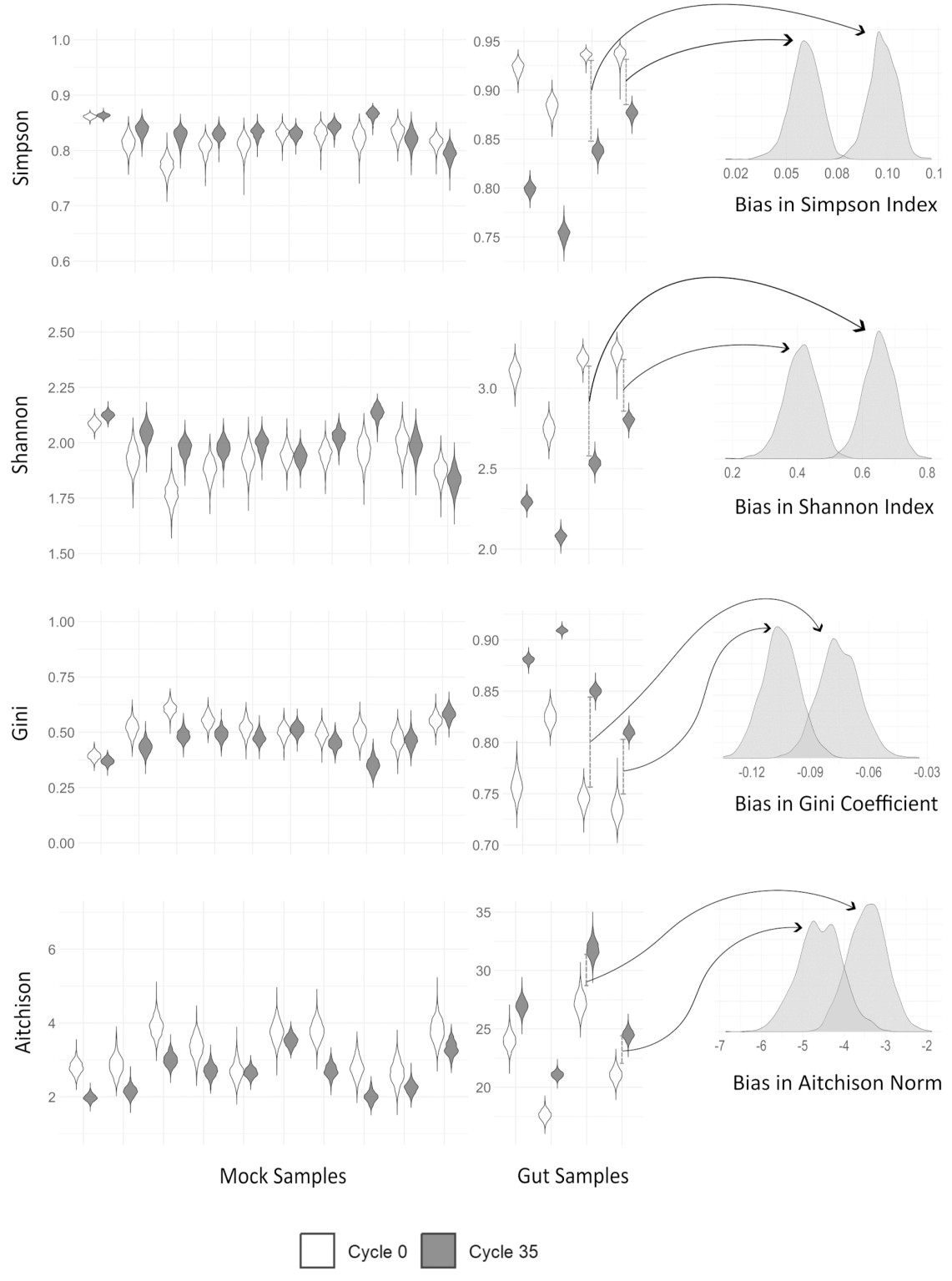

**Fig 1. $\alpha$-diversity metrics are impacted by PCR bias.** Violin plots show posterior distributions of four $\alpha$-diversity metrics (Shannon, Simpson, Gini, and Aitchison) for *in vitro* and *ex vivo* data, estimated before amplification ($0^{th}$ cycle) and after 35 PCR cycles using the model of Silverman et al. [8]. Density plots to the right show the posterior distributions of the PCR-induced bias (35-cycle value minus 0-cycle value) for two representative samples chosen because they exhibit the largest difference in PCR-induced bias among all samples. PCR introduces substantial, sample-specific shifts, indicating that comparisons of $\alpha$-diversity across groups may reflect technical artifacts rather than true biological differences.

this change as $\Delta R^2 = |R_{35}^2 - R_0^2|$, where $R_0^2$ and $R_{35}^2$ are ANOVA $R^2$ values at 0 and 35 PCR cycles, respectively. All four metrics showed substantial shifts: Simpson's index had a $\Delta R^2$ of 0.37, Shannon and Aitchison indices each reached 0.48, and the Gini coefficient exhibited the largest shift at 0.54. These results demonstrate that PCR bias could produce large apparent differences between groups, underscoring the need for caution when interpreting differential $\alpha$-diversity analyses.

To illustrate how PCR bias in $\alpha$-diversity depends on community composition, we used a simplified three-taxon system for visualization (Figs 2 and S3). This hypothetical example is not drawn from the mock data; rather, it demonstrates how the magnitude and direction of bias vary across the compositional simplex. Bias is minimal near the edges and vertices, where one or a few taxa dominate, and increases toward the interior, where communities are more even. In highly uneven communities, diversity metrics change little because small perturbations to minor taxa have limited impact. In contrast, even communities are far more susceptible–small shifts in relative abundance can substantially alter the balance among taxa. Although the specific location of the highest-bias region depends on which taxon is most over- or under-amplified, the overall pattern is consistent across the PCR-efficiency scenarios we examined: compositions that assign substantial weight to strongly biased taxa or that lie in moderately even regions of the simplex experience the largest distortions.

These visualizations help explain why some communities in the mock and gut datasets showed large shifts in differential $\alpha$-diversity analyses, whereas others were less affected (Fig 1). The mock communities were constructed in a way that naturally produces one or two dominant taxa [8]. As a result, these samples tend to occupy the low-bias, compositionally uneven regions of the simplex—where bias typically leads, for example, to overestimation of Shannon diversity but only

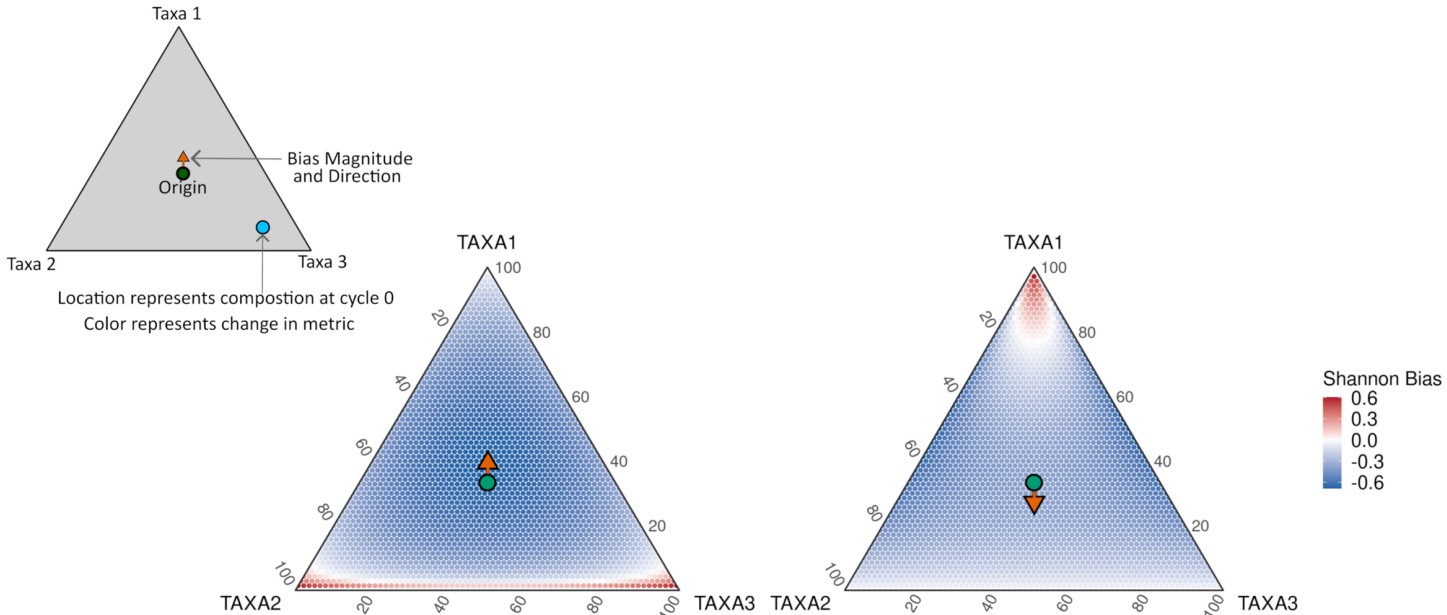

**Fig 2**. **Bias in Shannon index varies across the compositional space.** Ternary diagrams illustrate how PCR amplification alters Shannon diversity under two different bias scenarios. On the left the PCR bias preferentially amplifies TAXA 1 relative to the other two taxa, whereas on the right, TAXA 2 and 3 are preferentially amplified over TAXA 1. Each point represents an initial (pre-amplification) three-taxon composition, and its color indicates the change in Shannon diversity after 35 PCR cycles. The orange arrow shows the direction and magnitude of the PCR bias vector in composition space, and the green dot marks the unbiased origin. Even small differences in amplification efficiency produce highly non-uniform distortions, with the magnitude of bias depending strongly on the initial community composition.

by a modest amount. In contrast, several gut samples contain moderately abundant genera that Silverman et al. [8] identified as strongly over- or under-amplified (e.g., Bacteroides, Blautia, Faecalibacterium). Because these efficiency-sensitive taxa are neither rare nor overwhelmingly dominant, gut communities often lie in the high-bias, higher-richness/higher-evenness regions of the simplex and therefore exhibit substantially larger $\alpha$-diversity distortions after PCR.

### 2.4 Ecological $\beta$-diversity analyses are also impacted by PCR bias

$\beta$-diversity metrics quantify differences in community composition between samples and are widely used in microbiome research when $\alpha$-diversities alone cannot distinguish communities with similar richness and evenness but different taxonomic structures. Among commonly used indices, Weighted-UniFrac incorporates phylogenetic relationships by weighting abundance differences by branch length, whereas Bray-Curtis measures compositional dissimilarity based on relative abundances [25,26]. Because both metrics are perturbation sensitive, they may be affected by PCR bias. Formally, for a $\beta$-diversity metric $g : \mathbb{S}^D \times \mathbb{S}^D \to \mathbb{R}_+$, we define bias as:

$$\text{Bias} = g(\phi^{-1}(\alpha_1 + 35\,\beta), \phi^{-1}(\alpha_2 + 35\,\beta)) - g(\phi^{-1}(\alpha_1), \phi^{-1}(\alpha_2)),$$

where $\alpha_1$ and $\alpha_2$ are the log-ratio compositions of the two communities before amplification.

Using the Bayesian multinomial logistic model from Silverman et al. [8] we analyzed four artificial gut systems, each inoculated with a distinct human fecal sample and cultured ex vivo under conditions designed to mimic host physiology (see Sect 2.3). A parallel analysis of the mock community samples described in prior sections is provided in S4 Fig. We estimated pairwise Bray-Curtis and Weighted-UniFrac distances before amplification (0 cycles) and after 35 PCR cycles (Fig 3). PCR introduced substantial, sample-specific shifts in both metrics, with the magnitude of bias varying considerably across sample pairs. To further quantify these distortions, we examined the posterior distributions of the relative change in each pairwise distance of the same four artificial gut systems, revealing that PCR can either inflate or deflate $\beta$-diversity depending on the underlying community compositions being compared (S5 Fig). Formally, for a $\beta$-diversity metric $g : \mathbb{S}^D \times \mathbb{S}^D \to \mathbb{R}^+$, we define relative bias as:

$$\text{Relative Bias} = \frac{g(\phi^{-1}(\alpha_1 + 35\,\beta), \phi^{-1}(\alpha_2 + 35\,\beta)) - g(\phi^{-1}(\alpha_1), \phi^{-1}(\alpha_2))}{g(\phi^{-1}(\alpha_1), \phi^{-1}(\alpha_2))}$$

To evaluate how strongly PCR bias could affect downstream inference, we used the mock community data to test a worst-case scenario for differential $\beta$-diversity analyses. Because the original study lacked natural experimental groupings, we applied an optimization procedure to construct groupings that maximized the change in PERMANOVA $R^2$ across PCR cycles. We defined this change as $\Delta R^2 = |R^2_{35} - R^2_0|$, where $R^2_0$ and $R^2_{35}$ are PERMANOVA $R^2$ values at 0 and 35 cycles, respectively. The optimized configuration yielded modest but non-negligible shifts: $\Delta R^2 = 0.08$ for Bray-Curtis and 0.12 for Weighted-UniFrac. These results indicate that PCR bias can plausibly distort community-level comparisons.

Finally, to build intuition for why some community pairs are more affected than others, we visualized PCR-induced bias using a simplified three-taxon system (Figs 4 and S6). This hypothetical example, included for visualization only, illustrates how the magnitude of bias depends on where the two communities lie in the compositional simplex. Bias is smallest when the PCR-induced shift is orthogonal to the axis that characterizes the difference between the two communities. In contrast, Bias is largest when the shift impacts the relative abundance of the very taxa that distinguish the two communities. This helps explain the patterns in Figs 3 and S4. Gut samples can be distinguished by several moderately abundant taxa that Silverman et al. [8] identified as strongly over- or under-amplified (e.g., Bacteroides, Blautia, Faecalibacterium), placing these communities in high-bias regions of the simplex. In contrast, the mock communities are compositionally uneven and dominated by a small number of taxa whose amplification efficiencies are relatively similar; as a

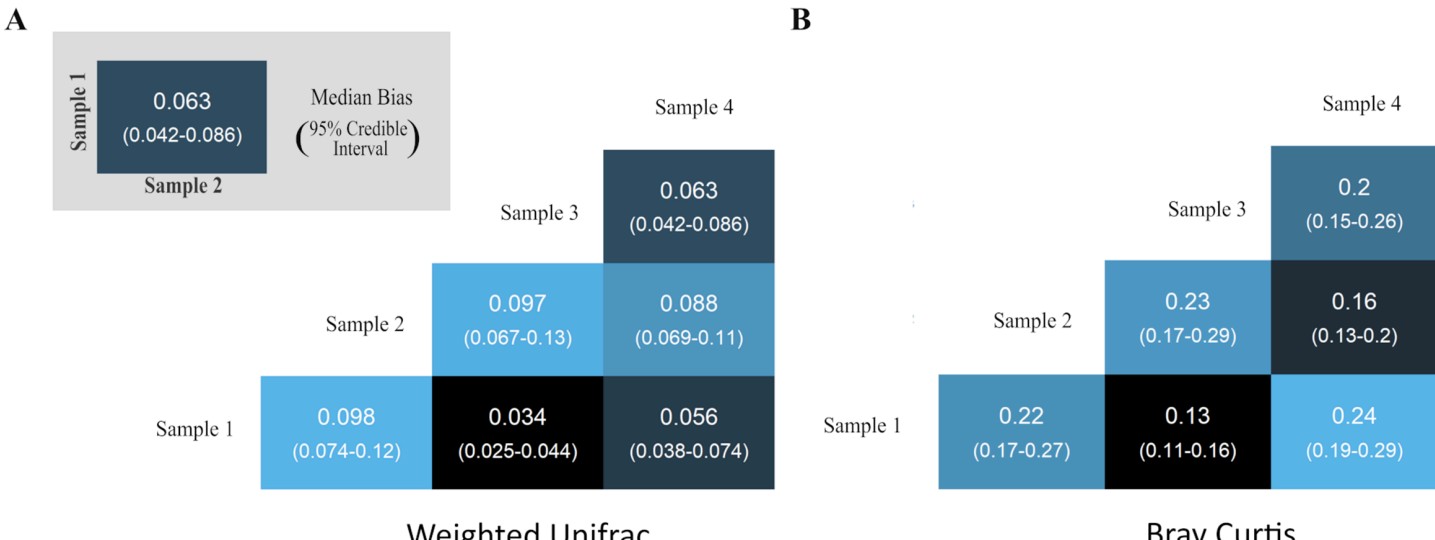

**Fig 3. PCR bias alters pairwise β-diversity estimates.** Heatmaps show the change in Weighted-UniFrac (A) and Bray-Curtis (B) distances between pairs of gut microbiome samples after 35 PCR cycles compared to the pre-amplification (0-cycle) values. Each tile shows the posterior median bias, with 95% credible intervals in parentheses. Variation across pairs indicates that PCR bias introduces systematic, sample-specific distortions in inter-sample dissimilarities.

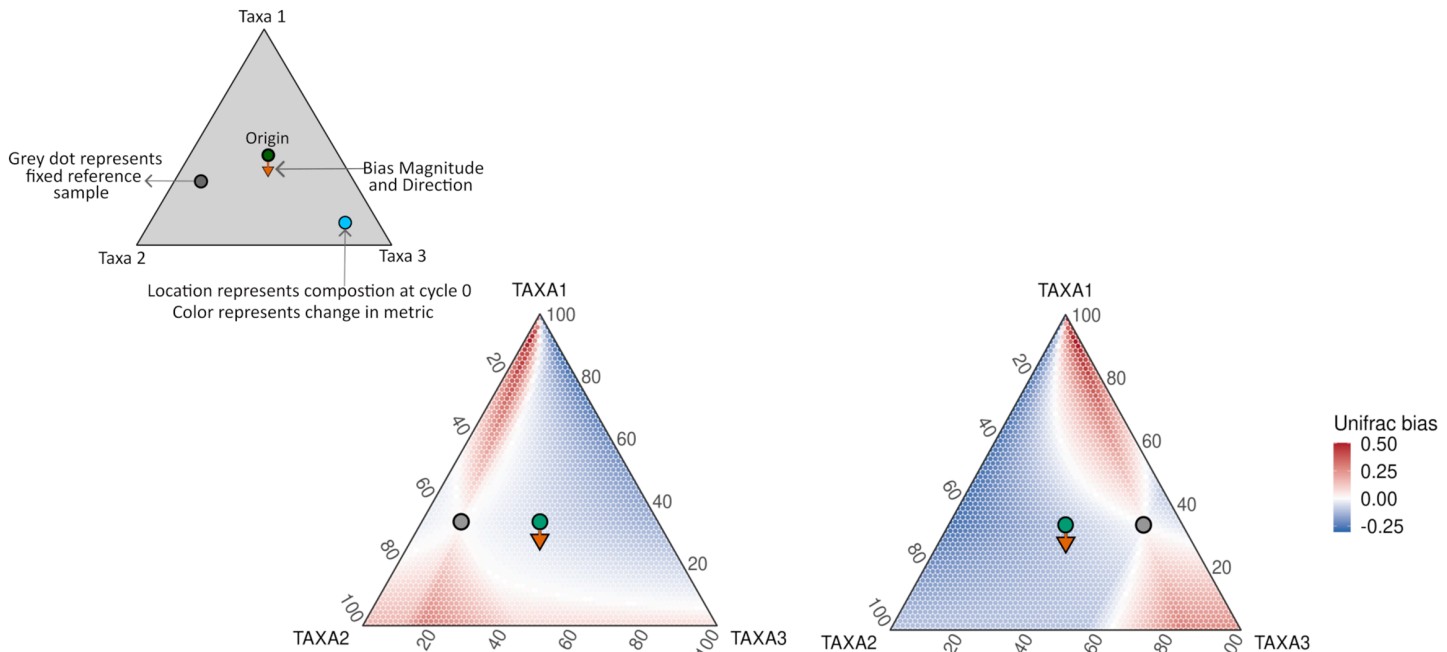

**Fig 4. Bias in Weighted-UniFrac depends on community composition.** Ternary diagrams show how Weighted-UniFrac bias varies as the second community moves across the compositional simplex relative to a fixed reference community, with PCR bias held constant. Color indicates the change in Weighted-UniFrac (35-cycle minus 0-cycle value). The orange arrow shows the direction and magnitude of PCR bias, the green dot marks the unbiased origin, and the grey circle marks the fixed reference.

result, they tend to fall in stable, low-bias regions across scenarios. This visualization therefore provides intuition for why PCR-induced bias varies so markedly between the mock and gut datasets.

## 2.5 Aitchison distance remains invariant to PCR bias

Unlike other $\beta$-diversity metrics, the Aitchison distance is unaffected by PCR bias because it is perturbation invariant, meaning it is unchanged by the additive shifts in log-ratio space introduced during amplification. For two communities $n_1$ and $n_2$, the Aitchison distance is defined as:

$$\delta_n = \sqrt{\sum_{d=1}^{D} \left(\eta_{dn_1} - \eta_{dn_2}\right)^2},$$

where $\eta_n = \phi(\pi_n)$ is the log-ratio transformed relative abundance vector for sample $n$. Intuitively, this distance measures how far apart two communities are in log-ratio space: a value of zero indicates identical compositions, whereas larger values reflect greater differences in the compositional structure of the communities. Because the distance is Euclidean in log-ratios, differences accumulate additively across log-ratios, making it straightforward to interpret which log-ratios contribute most to the separation.

As we prove in Corollary 2 (S1 Appendix), the additive perturbations introduced by PCR amplification cancel in this log-ratio representation, leaving the distance unchanged. This invariance has important practical implications. Unlike Bray-Curtis or Weighted UniFrac, which we show to be highly sensitive to PCR bias, the Aitchison distance provides a robust framework for $\beta$-diversity analysis. Because it remains stable even when amplification bias is present, the Aitchison distance helps ensure that observed differences between communities reflect genuine ecological or experimental variation rather than PCR-induced artifacts.

## 3 Discussion

While PCR is a routine step in microbiome profiling, differences in taxon-specific amplification efficiencies can introduce substantial bias in estimated community compositions. Primer mismatch effects occur during the initial amplification cycles, but multiple studies have shown that non–primer-mismatch (NPM) sources of bias dominate PCR bias overall [8,11,16]. Here we extended prior studies of NPM sources of PCR bias (called PCR bias for brevity) to systematically evaluate how these biases affect ecological diversity analyses.

Our results show that PCR bias alters both $\alpha$- and $\beta$-diversity estimates, with the magnitude and direction of distortion depending strongly on underlying community composition. For $\alpha$-diversity, bias is largest in communities with even taxonomic structure, where small perturbations in relative abundance produce large changes in Shannon, Simpson, Gini, or Aitchison-based metrics. At present, we are not aware of any commonly used $\alpha$-diversity metric that is perturbation invariant, and whether such a metric can be developed remains an open question. For $\beta$-diversity, bias is minimal when the PCR-induced perturbation is largely orthogonal to the differences between communities, but can be substantial when it aligns with dominant taxonomic or phylogenetic axes of variation. In these cases, PCR bias can obscure or exaggerate apparent ecological differences between groups. Using ANOVA ($\alpha$-diversity) and PERMANOVA ($\beta$-diversity), we further show that bias can amplify or suppress group-level signals, potentially affecting ordination, clustering, and differential-diversity analyses.

Because the mock community dataset lacked natural experimental groupings, we constructed artificial groupings using an optimization procedure designed to maximize changes in statistical signal across PCR cycles. This produces a worst-case scenario: it identifies sample configurations that are maximally aligned with the directions of variation imposed by amplification bias. Real biological groupings are unlikely to align this strongly, but partial alignment could still produce

noticeable inflation or attenuation of diversity-based signals. The optimized groupings therefore serve as a useful interpretive benchmark, clarifying the specific geometric circumstances under which PCR bias exerts its strongest effects and providing context for evaluating the magnitude of bias observed in empirical studies. Further research using empirical datasets is needed to determine how representative these cases are and to better understand the extent to which PCR bias alters diversity-based conclusions in practice.

In contrast to these diversity-based estimands, which can be highly sensitive to the geometry of the underlying compositions, some quantities are inherently robust to PCR-induced perturbations. When all samples are amplified under the same protocol, the multiplicative model of PCR bias implies that taxon-specific efficiencies act uniformly across samples, producing a constant additive shift in log-ratio space. As we show formally, perturbation-invariant estimands–such as differential log-ratios and the Aitchison distance–are unaffected by such shifts. Intuitively, on the simplex (as in Fig 2), perturbation invariance means that every initial composition experiences zero bias; e.g., Figs 2 or 4 would appear completely white. Such estimands therefore remain stable regardless of the community composition, in stark contrast to standard $\alpha$- and $\beta$-diversity measures whose sensitivity to PCR bias varies across the simplex.

Given these findings, we recommend using compositional metrics based on log-ratio transformations—such as the Aitchison distance—whenever possible for analyses of PCR-amplified microbiome data. These methods are perturbation invariant and therefore unaffected by PCR bias. However, the choice of metric should ultimately be guided by the underlying biological question. If the question depends on a perturbation-sensitive metric, then efforts should be taken to quantify and mitigate PCR bias to ensure valid inference. In particular, we recommend performing calibration experiments (e.g., sequencing the same community with different numbers of PCR cycles) to estimate community-specific amplification bias and adjust diversity measures following the approaches described here and in Silverman et al. [8]. When such mitigation is not possible, results from perturbation-sensitive diversity analyses should be interpreted with caution, both in their absolute values and in between-sample comparisons.

These results also have implications for differential abundance analyses. Many widely used DA tools (e.g., DESeq2 and edgeR) operate directly on count data or normalized counts, making them perturbation sensitive. In contrast, methods based on log-ratio coordinates–including those implemented in fido–target perturbation-invariant estimands and are therefore unaffected by PCR-induced shifts in relative abundance. Our findings do not prescribe a single "best" method, but they do suggest that log-ratio–based approaches provide greater robustness when PCR bias is present or cannot be experimentally calibrated.

Although our analyses focus on 16S rRNA gene sequencing, the framework developed here likely extends to other sequencing applications in which PCR amplification is used during library preparation. Many shotgun metagenomic and viromic workflows employ PCR–especially for low-input samples–and targeted amplicon, immune repertoire, and RNA-seq protocols rely on amplification steps that can introduce sequence- or taxon-specific distortions in observed abundance profiles. While we are not aware of published studies showing that PCR bias alone has produced incorrect biological conclusions in these contexts, such outcomes are plausible given the ubiquity of PCR and the sensitivity of many diversity metrics to amplification-induced perturbations. Applying perturbation-invariant measures or bias-correction methods in these broader settings may help identify and mitigate such effects. More generally, the distinction between perturbation-sensitive and perturbation-invariant estimands provides a unifying framework for improving robustness in sequencing-based inference and may prove valuable for PCR-dependent analyses across metagenomics, transcriptomics, and immune repertoire profiling.

Our analysis has focused on continuous functions of relative abundances and has not considered presence/absence-based metrics (e.g., Jaccard, unweighted UniFrac), which are more sensitive to detection thresholds and sequencing noise. Although PCR bias can theoretically push rare taxa below detection, modeling such effects requires explicitly accounting for detection variability and is beyond the scope of this study. Future work should assess how PCR bias interacts with presence/absence metrics and evaluate its impact across a broader range of experimental settings. Independent

validation data, such as technical replicates or mock communities, will be essential for determining the operational relevance of these findings in real-world microbiome studies. As more datasets become available, our understanding of the ecological consequences of PCR bias will continue to improve.

## 4 Methods

### 4.1 PCR bias model with covariates

We modeled PCR bias using the Bayesian multinomial logistic-normal framework introduced by Silverman et al. [8], which extends the standard exponential amplification model to accommodate multiple taxa, technical noise, and structured sample covariates. The model is specified as:

$$\mathbf{Y}_n \sim \text{Multinomial}(\pi_n)$$
$$\pi_n = \phi^{-1}(\eta_n)$$
$$\eta_n \sim \mathcal{N}(\Lambda \mathbf{X}_n, \Sigma)$$

where $Y_n$ is the vector of sequencing read counts for sample $n$, $\pi_n$ is the corresponding vector of relative abundances, and $\eta_n = \phi(\pi_n)$ denotes the log-ratio transformed abundance. The matrix $\Lambda \in \mathbb{R}^{(D-1) \times p}$ contains regression coefficients, $\Sigma \in \mathbb{R}^{(D-1) \times (D-1)}$ is the residual covariance, and $X_n \in \mathbb{R}^p$ is a covariate vector specifying sample-level features.

This formulation allows for flexible modeling of complex experimental designs. For example, to jointly model samples from two biological communities with varying numbers of PCR cycles, we can construct a design matrix $X \in \mathbb{R}^{p \times N}$, where each column $X_n$ includes a one-hot encoding of community membership and a numeric variable for PCR cycle count:

$$X_n = \begin{bmatrix} I_{C_1}(n) \\ I_{C_2}(n) \\ x_n \end{bmatrix},$$

with $I_{C_i}(n) = 1$ if sample $n$ belongs to community $C_i$ (and zero otherwise), and $x_n$ denoting the number of PCR cycles applied. In this case, the coefficient matrix $\Lambda$ takes the form:

$$\Lambda = \begin{bmatrix} \alpha_1^{(1)} & \alpha_1^{(2)} & \beta_1 \\ \vdots & \vdots & \vdots \\ \alpha_{D-1}^{(1)} & \alpha_{D-1}^{(2)} & \beta_{D-1} \end{bmatrix},$$

where $\alpha_l^{(i)}$ is the baseline (cycle-0) log-ratio abundance of the $l^{\text{th}}$ contrast in community $C_i$, and $\beta_l$ represents the per-cycle PCR amplification bias on that log-ratio. This structure enables simultaneous inference of both biological variation and amplification bias, even when only some samples vary in PCR cycle number.

All model fitting was performed in R using the fido package [18]. *fido* implements scalable Bayesian inference for multinomial logistic-normal models using the collapse–uncollapse sampler introduced in Silverman et al. [18]. In brief, *fido* exploits a marginal-Laplace inference strategy: (i) it first *collapses* the model by analytically marginalizing over the regression parameters $\alpha$, $\beta$, and the covariance $\Sigma$; (ii) it then approximates the marginal posterior of the latent log-ratio variables using a *Laplace approximation*; and (iii) it *uncollapses* by conditioning on samples from this Laplace approximation to generate posterior samples of $\alpha$, $\beta$, and $\Sigma$. This approach avoids high-dimensional MCMC over latent variables and enables fast and accurate posterior inference [18].

## 4.2 Datasets and preprocessing

We analyzed two publicly available datasets originally described by Silverman et al. [8]: ten in vitro mock communities composed of known proportions of ten bacterial isolates and four ex vivo human gut microbiota samples derived from distinct artificial gut systems. Each sample was split into three replicates, with each replicate subjected to a different number of PCR cycles prior to sequencing, enabling estimation of all model parameters. All analyses were conducted using the same preprocessed datasets and parameter settings as in the original study. We re-ran the publicly available `fido` pipeline without modification to estimate posterior distributions of $\alpha_n$ (true relative abundances for each community at cycle 0) and $\beta$ (taxon-specific relative amplification efficiencies).

## 4.3 Diversity metrics and bias estimation

We evaluated the effect of PCR bias on commonly used ecological diversity measures. Posterior samples from the fitted model were used to estimate taxon relative abundances at 0 and 35 PCR cycles, which served as inputs for downstream calculations.

$\alpha$**-diversity.** We analyzed four $\alpha$-diversity metrics: Shannon index, Simpson index, Gini coefficient, and the Aitchison norm. Shannon, Simpson, and Gini indices were computed using custom `R` functions. The Aitchison norm was calculated as the Euclidean norm of centered log-ratio (CLR) transformed abundance vectors. PCR-induced bias for each metric was defined as:

$$\text{Bias}_n = f(\phi^{-1}(\alpha_n + 35\beta)) - f(\phi^{-1}(\alpha_n)).$$

$\beta$**-diversity.** We evaluated three $\beta$-diversity metrics: Bray-Curtis dissimilarity, Weighted UniFrac, and Aitchison distance. Bray-Curtis dissimilarities were calculated using the `vegdist()` function in the `vegan` package. Weighted UniFrac distances were computed using the `UniFrac()` function in the `phyloseq` package with a pruned `greengenes2` phylogenetic tree. Aitchison distances were calculated as Euclidean distances in Centered Log-Ratio (CLR) space. Bias was defined analogously to $\alpha$-diversity metrics:

$$\text{Bias}_{n_1,n_2} = g(\phi^{-1}(\alpha_{n_1} + 35\beta), \phi^{-1}(\alpha_{n_2} + 35\beta)) - g(\phi^{-1}(\alpha_{n_1}), \phi^{-1}(\alpha_{n_2})).$$

## 4.4 Statistical analyses

**Alpha diversity.** To quantify the effect of PCR bias on group-level diversity comparisons, we computed mean diversity values per sample at both cycle 0 and cycle 35 and evaluated group differences using ANOVA. A genetic algorithm was applied to identify binary groupings of samples that maximized the absolute change in ANOVA $R^2$ across PCR cycles:

$$\Delta R^2 = |R_{35}^2 - R_0^2|.$$

**Beta diversity.** To assess changes in community structure, we performed PERMANOVA (`adonis2()` in `vegan`) on each $\beta$-diversity distance matrix. Particle swarm optimization (`psoptim()` in `R`) was used to find the grouping of samples that maximized changes in PERMANOVA $R^2$ across PCR cycles, providing a worst-case estimate of how much amplification could distort community-level comparisons.

## 4.5 Visualization of composition-dependent bias

To build intuition for composition-dependent effects, we simulated PCR bias in a simplified three-taxon system. Bias was evaluated across a grid of compositions in the 3-part simplex, holding the amplification bias vector constant. This visualization highlights how bias depends on the position of communities in the simplex and explains variability observed in the mock community analyses.

## Supporting information

**S1 Appendix Supplementary Theorems and Proofs.** This supplement contains theorems establishing perturbation-invariant estimands, which remain robust under PCR-induced bias. It also includes corollaries on differential log-ratio abundances and the Aitchison distance, providing theoretical support for compositional analyses in microbiome studies. (PDF)

**S1 Fig. PCR bias in human gut and mock communities.** Bias was quantified as the $\log_2$-ratio of each taxon's estimated relative abundance after 35 PCR cycles (cycle 35) to its estimated pre-amplification abundance (cycle 0) for (A) human gut microbiota and (B) mock communities from Silverman et al. [8]. A $\log_2$ value of 2 indicates a fourfold overestimation due to PCR, whereas -2 indicates a fourfold underestimation. Points show posterior medians, and bars denote 95% credible intervals. Taxa with credible intervals excluding zero (statistically credible bias) are highlighted in black. (TIFF)

**S2 Fig. Relative bias in $\alpha$-diversity metrics.** Violin plots show posterior distributions of sample-specific relative bias for four $\alpha$-diversity metrics (Simpson, Shannon, Gini, Aitchison) across gut and mock communities. The shapes of the distributions highlight that the magnitude and direction of PCR bias vary systematically across metrics and differ between controlled mock communities and complex gut communities. Together, these patterns demonstrate that PCR amplification can introduce non-uniform distortions that depend both on the diversity metric and on underlying community structure. (TIFF)

**S3 Fig. Composition-dependent bias in Simpson, Gini, and Aitchison diversity.** Ternary diagrams show PCR-induced changes in (A) Simpson diversity, (B) Gini coefficient, and (C) Aitchison norm after 35 cycles under two different bias vectors (orange arrow). On the left the PCR bias preferentially amplifies TAXA 1 relative to the other two taxa, whereas on the right, TAXA 2 and 3 are preferentially amplified over TAXA 1. Each point represents an initial pre-amplification composition in a three-part simplex, colored by the change in the corresponding metric after PCR. The green dot marks the unbiased origin, and the orange arrow indicates the direction and magnitude of the bias vector. Even small differences in amplification efficiency produce highly non-uniform distortions, with bias magnitude varying systematically across the compositional space. (TIFF)

**S4 Fig. Pairwise $\beta$-diversity bias in mock communities.** Heatmaps show changes in (A) Weighted UniFrac and (B) Bray-Curtis distances between pairs of mock community samples after 35 PCR cycles relative to their pre-amplification (cycle 0) values. Each tile shows the posterior median change, with 95% credible intervals in parentheses. Rows and columns correspond to sample indices. Variation in bias across pairs indicates that PCR amplification introduces systematic and sample-specific distortions in inter-sample dissimilarities. (TIFF)

**S5 Fig. Relative bias in $\beta$-diversity distances for gut communities.** Plots show posterior distributions of pairwise relative bias for Bray-Curtis and weighted UniFrac distances, summarizing how PCR alters dissimilarity between samples. The distributions reveal substantial variation in magnitude of distortion across sample pairs, reflecting the dependence of PCR-induced changes on the underlying community compositions being compared. These patterns indicate that PCR amplification can introduce non-uniform, comparison-specific distortions in $\beta$-diversity estimates. (TIFF)

**S6 Fig. Composition-dependent bias in Bray-Curtis dissimilarity.** Change in Bray-Curtis distance is shown as the second community varies across the simplex relative to a fixed reference composition, with PCR bias held constant. Colors indicate the difference between the Bray-Curtis value at 35 PCR cycles and its true pre-amplification value.

The green dot marks the unbiased origin, the orange arrow indicates the direction of PCR bias, and the grey circle marks the fixed reference community.

(TIFF)

## Acknowledgments

The authors would like to thank Dr. Rachel Silverman for her manuscript comments.

## Author contributions

**Conceptualization:** Justin D. Silverman.

**Data curation:** Justin D. Silverman.

**Formal analysis:** Dharmik R. Rathod.

**Funding acquisition:** Justin D. Silverman.

**Investigation:** Dharmik R. Rathod.

**Methodology:** Justin D. Silverman.

**Project administration:** Justin D. Silverman.

**Supervision:** Justin D. Silverman.

**Validation:** Dharmik R. Rathod.

**Visualization:** Dharmik R. Rathod.

**Writing – original draft:** Dharmik R. Rathod, Justin D. Silverman.

**Writing – review & editing:** Justin D. Silverman.

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
