## [Decision Letter · Decision Letter 0]

3 Nov 2025

PCOMPBIOL-D-25-01547

PCR Bias Impacts Microbiome Ecological Analyses

PLOS Computational Biology

Dear Dr. Silverman,

Thank you for submitting your manuscript to PLOS Computational Biology. After careful consideration, we feel that it has merit but does not fully meet PLOS Computational Biology's publication criteria as it currently stands. Therefore, we invite you to submit a revised version of the manuscript that addresses the points raised during the review process.

Please submit your revised manuscript within 30 days Jan 03 2026 11:59PM. If you will need more time than this to complete your revisions, please reply to this message or contact the journal office at ploscompbiol@plos.org. Please include the following items when submitting your revised manuscript:

We look forward to receiving your revised manuscript.

Kind regards,

Nic Vega, Ph.D.

Academic Editor

PLOS Computational Biology

Virginia Pitzer

Editor-in-Chief

PLOS Computational Biology

**Additional Editor Comments (if provided):**

The reviewers' comments are essentially minor. Both reviewers indicate a number of cases where better connection between the theory and real data would help to both clarify the theory and to emphasize the impact of the work.

**Journal Requirements:**

**Reviewers' comments:**

Reviewer's Responses to Questions

**Comments to the Authors:**

Reviewer #1: Comments attached

Reviewer #2: This is an excellent, well-executed, and highly significant manuscript that makes a substantial contribution to the field of microbiome bioinformatics. The research is rigorous, the concept of "perturbation invariance" is a powerful and elegant theoretical framework that effectively explains and predicts the effects of PCR bias. The study successfully bridges a critical knowledge gap by moving beyond the established impact on relative abundances to provide a systematic evaluation of ecological diversity metrics. The conclusions are well-supported by both theoretical proofs and empirical data, and the practical recommendations are clear and actionable.

1. The use of an optimization procedure to create groupings that maximize delta R^2 is clever for demonstrating potential impact. However, it risks being perceived as a strawman argument, as reviewers may question how often such a worst-case scenario occurs in real studies. Please clearly state that this is a "proof-of-concept" or "worst-case" analysis designed to demonstrate the maximum plausible effect of PCR bias, not necessarily a common one. Discuss factors that would make such a severe distortion more or less likely in real data (e.g., how aligned the true biological signal is with the PCR bias vector). This preempts a potential criticism.

2. Figures 2 & 4 (Simplex Plots): These are excellent for intuition but currently feel somewhat disconnected from the real data analysis. The manuscript would be stronger if you could directly link the patterns in these theoretical simplices to specific sample pairs from your mock/gut data. For example, could you point to a pair in Figure S3 and explain its high/low bias using the principles illustrated in Figure 4?

3. Figure 1: The violin plots effectively show that bias exists and is sample-specific. However, the "two representative samples" for the bias density plots are not identified in the main text. Why were these chosen? Label them (e.g., "Mock Community A," "Gut Sample B") and briefly justify their selection (e.g., "one showing large positive bias, one showing large negative bias").

4. You correctly state that differential log-ratios (a perturbation-invariant estimand) are unaffected by PCR bias. However, many popular differential abundance tools (e.g., DESeq2, edgeR, ANCOM-BC) do not explicitly use this framework and are applied to raw counts. There is an ongoing debate in the field about the best methods.

You may briefly discuss the implication of your finding for the choice of differential abundance methods. Does your work implicitly support the use of log-ratio-based methods (like those built on fido) over others when PCR bias is a concern? A sentence or two would add significant translational impact.

5. Please formalize the definition of "Perturbation-Invariant Estimand". Before giving the formal definition, state in plain language: "An estimand is perturbation-invariant if its value does not change when every log-ratio in the composition is shifted by the same constant value." Then, explicitly state that PCR bias, under a consistent protocol, acts as precisely such a constant shift in log-ratio space. This builds a stronger bridge between the intuition and the math.

6. The reference list is generally good but has a few minor formatting inconsistencies (e.g., journal names in italics vs. not, use of "et al.").

**Have the authors made all data and (if applicable) computational code underlying the findings in their manuscript fully available?**

Reviewer #1: Yes

Reviewer #2: Yes

PLOS authors have the option to publish the peer review history of their article (what does this mean?). If published, this will include your full peer review and any attached files.

Reviewer #1: **Yes:** Alan DenAdel

Reviewer #2: **Yes:** Shi Huang

**Figure resubmission:**
---

## [Editor Report · Decision Letter 1]

9 Jan 2026

Dear Dr. Silverman,

We are pleased to inform you that your manuscript 'PCR Bias Impacts Microbiome Ecological Analyses' has been provisionally accepted for publication in PLOS Computational Biology.

Best regards,

Nic Vega, Ph.D.

Academic Editor

PLOS Computational Biology

Virginia Pitzer

Editor-in-Chief

PLOS Computational Biology

---

## [Editor Report · Acceptance letter]

PCOMPBIOL-D-25-01547R1

PCR Bias Impacts Microbiome Ecological Analyses

Dear Dr Silverman,

I am pleased to inform you that your manuscript has been formally accepted for publication in PLOS Computational Biology. Your manuscript is now with our production department and you will be notified of the publication date in due course.

With kind regards,

Zsofia Freund
